# Experimental Infection of Brazilian Free-Tailed Bats (*Tadarida brasiliensis*) with Two Strains of SARS-CoV-2

**DOI:** 10.3390/v14081809

**Published:** 2022-08-18

**Authors:** Angela M. Bosco-Lauth, Stephanie M. Porter, Karen A. Fox, Mary E. Wood, Daniel Neubaum, Marissa Quilici

**Affiliations:** 1Department of Biomedical Sciences, Colorado State University, Fort Collins, CO 80523, USA; 2U.S. Department of Agriculture, Animal and Plant Health Inspection Service, Wildlife Services, National Wildlife Research Center, Fort Collins, CO 80523, USA; 3Colorado Parks and Wildlife, Fort Collins, CO 80523, USA

**Keywords:** Brazilian free-tailed bat, *Tadarida brasiliensis*, SARS-CoV-2, infection

## Abstract

Severe acute respiratory syndrome coronavirus 2 (SARS-CoV-2) is presumed to have originated from wildlife and shares homology with other bat coronaviruses. Determining the susceptibility of North American bat species to SARS-CoV-2 is of utmost importance for making decisions regarding wildlife management, public health, and conservation. In this study, Brazilian free-tailed bats (*Tadarida brasiliensis)* were experimentally infected with two strains of SARS-CoV-2 (parental WA01 and Delta variant), evaluated for clinical disease, sampled for viral shedding and antibody production, and analyzed for pathology. None of the bats (n = 18) developed clinical disease associated with infection, shed infectious virus, or developed histopathological lesions associated with SARS-CoV-2 infection. All bats had low levels of viral RNA in oral swabs, six bats had low levels of viral RNA present in the lungs during acute infection, and one of the four bats that were maintained until 28 days post-infection developed a neutralizing antibody response. These findings suggest that Brazilian free-tailed bats are permissive to infection by SARS-CoV-2, but they are unlikely to contribute to environmental maintenance or transmission.

## 1. Introduction

The virus responsible for the coronavirus disease 2019 (COVID-19) pandemic, severe acute respiratory syndrome 2 (SARS-CoV-2), possibly originated from wildlife, and the most closely related viruses are currently found in Asian Rhinolophus bats and pangolins [1,2,3,4,5,6,7]. SARS-CoV-2 has been shown to have a broad host range, with natural infections occurring in a variety of wildlife species in addition to humans and domestic animals [8]. Despite this, there have not been any identified instances of natural SARS-CoV-2 infection or disease in wild bats. The role of bats in starting, maintaining, or extending the SARS-CoV-2 pandemic is largely unknown, and as such, identifying bats that are susceptible to infection with the virus will help to determine which bat species to monitor closely as potential reservoirs for disease. Furthermore, many bat populations are already experiencing declines due to anthropogenic change and other pathogens [9]; understanding the threat posed by SARS-CoV-2 is critical for conservation management of at-risk species.

Of the more than 1400 species of bats worldwide, only two species have been experimentally infected with SARS-CoV-2 at the time of this publication: Egyptian fruit bats (*Rousettus aegyptiacus)* from the Old World, and big brown bats (*Eptesicus fuscus)* from the New World [10,11]. No evidence of infection was observed in the big brown bats, while the fruit bats appeared moderately permissive to infection, with low-level shedding detected by rt-PCR and no evidence of clinical disease. In a survey of wild Egyptian fruit bats, there was no evidence of circulating SARS-CoV-2 in 200 bats sampled in Egypt during the pandemic [12]. While these studies provide promising information regarding the lack of involvement of these two species, there are still many unanswered questions surrounding the role of bats in the COVID-19 pandemic.

This study evaluated Brazilian free-tailed bats (*Tadarida brasiliensis)* for susceptibility to SARS-CoV-2 infection. Brazilian free-tailed bats are a widespread insectivorous migratory bat species found throughout the South and Western U.S., Central and South America. They are one of the most abundant bat species in this region of the U.S. and can roost in communities of up to 15 million individuals [13]. Their larger roosting sites, such as the maternal colonies found near San Antonio, Texas, provide a considerable tourism draw. The abundance of these bats, in addition to their relative proximity to humans, makes them highly likely to encounter humans, some of whom could be infected with SARS-CoV-2. Furthermore, betacoronaviruses have been detected in the broad-eared bat (*Nyctinomops laticaudatus*) in Central America, and based on ACE2 orthologue receptor usage, *T. brasiliensis* have a medium to high likelihood of susceptibility to SARS-CoV-2 [14,15]. Brazilian free-tailed bats are known to cohabitate with other bat species, thereby increasing the risk of spreading viruses between species. Thus, it is important to determine how these bats respond to exposure, with particular emphasis on clinical disease and potential for viral shedding. The experimental infections described herein provide insight into how Brazilian free-tailed bats respond to exposure to SARS-CoV-2.

## 2. Materials and Methods

### 2.1. Capture of Bats

Free-ranging Brazilian free-tailed bats were captured from a maternity colony during late August 2021 while exiting a bridge in Grand Junction, CO, using a handheld H-net [16]. COVID-19-vaccinated personnel wore N95 masks and changed gloves between handling individuals. Bats were placed in individual cloth holding bags to await processing. Each bat was examined to estimate age class and determine sex before being placed in a temporary enclosure for transport to Colorado State University’s Animal Disease Laboratory (ADL), an Animal Biosafety Level 3 (ABSL3) facility. Bats were group housed in plastic tubs (18″ × 24″ × 18″) and fed a diet consisting of meal worms with ad libitum access to water. Bats were acclimated to captivity for several weeks, including transitioning to a diet of mealworms, which involved hand-feeding bats daily until they were observed eating on their own. Animals were captured and held in accordance with institutional ACUC procedures and with appropriate permitting.

### 2.2. Virus

SARS-CoV-2 virus strains WA01/2020WY96 and Delta hCoV-19/USA/MD-HP05647/2021 were obtained from BEI Resources (Manassas, VA, USA), passaged twice in Vero E6 cells and stocks frozen at −80 °C in Dulbecco’s Modified Eagle Medium (DMEM) with 5% fetal bovine serum and antibiotics. Virus stock was titrated on Vero cells using standard double overlay plaque assay [17] and plaques were counted 48 and 72 h later to determine plaque-forming units (pfu) per mL.

### 2.3. SARS-CoV-2 Challenge

Prior to challenge, oral swabs were collected, and all bats were confirmed negative for SARS-CoV-2 shedding. All bats included in the challenge study (n = 18) were females, 1 was a juvenile and the other 17 were adults. Six bats (four adult females, one juvenile female and one juvenile male) either died or were euthanized prior to challenge and were included as a non-infected control bats for histopathological comparison. Bats were challenged with either WA01 or Delta variant (n = 9 per strain) via intranasal inoculation. Bats were manually restrained and 20 µL undilute stock virus was inoculated dropwise into the nares via pipette. Virus back-titration was performed on Vero cells immediately following inoculation, confirming that animals received between 3.5 and 3.6 log_10_ pfu of SARS-CoV-2 WA01 and Delta, respectively.

### 2.4. Sample Collection

Following infection, oral swabs were collected into BA-1 viral transport media (Tris-buffered MEM containing 5% bovine serum albumin and antibiotics) daily for virus isolation through 7 days post-infection (dpi) and bats were monitored for clinical signs of infection daily throughout the course of the study. Up to 3 bats were harvested at days 3, 7 or 28 and necropsied; several bats died due to ill thrift in captivity in the days following infection; these were also necropsied. Nasal turbinates and lungs were collected into viral transport media, a terminal blood sample was collected and whole bats were fixed in formalin.

### 2.5. Viral Assays

Virus isolation was performed on all oral swab, lung and turbinate tissue samples via double overlay plaque assay on Vero cells, as previously described [17]. Briefly, 6-well plates with confluent monolayers of cells were washed once with PBS and inoculated with 100 µL of serial 10-fold dilutions of samples, incubated for 1 h at 37 °C, and overlaid with 0.5% agarose in MEM containing 2% fetal bovine serum and antibiotics/antifungal agents. A second overlay with neutral red dye was added at 24 h (WA01) and 48 h (Delta), and plaques were counted at 48 or 72 h, respectively. Viral titers were reported as the log_10_ pfu per swab (oral) or per gram (tissue).

### 2.6. Serology

Plaque reduction neutralization assays (PRNT) were performed as previously described [17]. Serum was heat inactivated for 30 min at 56 °C, and two-fold dilutions prepared in BA-1 (Tris-buffered MEM containing 1% bovine serum albumin) starting at a 1:10 dilution were aliquoted onto 96-well plates. An equal volume of virus was added to the serum dilutions and incubated for 1 h at 37 °C. Following incubation, serum–virus mixtures were plated onto Vero monolayers as described for virus isolation assays. Antibody titers were recorded as the reciprocal of the highest dilution in which >90% of virus was neutralized.

### 2.7. qRT-PCR

Lung homogenates and oral swabs were subjected to RNA extractions performed per the manufacturer’s instructions using Qiagen QiaAmp™ Viral RNA mini kits. RT-PCR was performed as recommended using the E_Sarbeco primer probe sequence as described by Corman and colleagues [18], and the Qiagen One-Step qRT-PCR™ kit with the following modification: the initial reverse transcription was at 50 °C. RNA standards for PCR were obtained from BEI Resources (Manassas, VA, USA).

### 2.8. Histopathology

Representative tissues were preserved in 10% neutral buffered formalin. Tissues included nasal turbinates, upper trachea, lower trachea, lung, stomach, small intestine, pancreas, colon, liver, mesenteric lymph node, spleen, thymus (in juvenile bats), heart, kidney and brain. Tissues were embedded in paraffin blocks, sectioned at 5–8 µm, adhered to glass slides, stained with hematoxylin and eosin and examined microscopically.

### 2.9. Immunohistochemistry

Bat tissue sections were fixed onto charged slides and deparaffinized. The sections were submerged in Dako Target Retrieval (Agilent Technologies Inc., Santa Clara, CA, USA) solution heated to 90 °C for 30 min, then allowed to cool in the solution for 30 min until they reached room temperature (RT). Cooled sections were rinsed with water and covered with 0.3% hydrogen peroxide for 10 min at RT. This was followed by a wash using TRIS-buffered saline with 0.1% Tween-20 (TBST) before they were covered with 0.15 M glycine in PBS for 15 min at RT, rinsed again and blocked with a solution of 1% bovine serum albumin and 10% fetal bovine serum in TBST for 30 min at RT. Blocking solution was discarded and the sections were covered overnight at 4 °C with rabbit anti-SARS IgG diluted at 1:400. The primary antibody (SARS Nucleocapsid Protein Antibody, Novus Biologicals, Centennial, CO, USA) was washed with TBST, and sections were then covered with goat anti-rabbit IgG diluted at 1:200 for 30 min, washed with TBST and covered with horseradish peroxidase/avidin solution for 30 min at RT. Bound antibody was visualized with DAB solution applied to the tissue for ~3–4 min at RT, washed with water, counterstained with hematoxylin for ~2 min and washed again. The stained sections were left to dry overnight, mounted and examined using a microscope.

## 3. Results

### 3.1. Virology, Clinical Disease and Serology

Throughout the course of the experimental infection, none of the bats shed infectious virus orally or had infectious virus in lungs or turbinates at the time of necropsy. Several bats adapted poorly to captivity, with low appetite and depressed activity. These bats either died or were euthanized before challenge (n = 6) and were used as control bats; several others died or were euthanized between days 2 and 5 post-infection, with evidence of bacteremia and parasitism (n = 4 for WA01 strain and n = 2 for Delta strain), likely secondary to stress and malnutrition. These bats were subjected to the same analyses, and none were shedding infectious virus orally or via the lungs at the time of death. Oral swabs collected between 1–7 DPI contained SARS-CoV-2 viral RNA at nearly every collection for all bats, regardless of inoculating strain (Table 1). Cycle threshold (CT) values were greater than 30 for all bats at all timepoints, except two bats at 1 DPI whose CT scores were 27 and 29 (bat 3 and 9, respectively) (Appendix A). The lack of detection of infectious virus indicates that the level of shedding was below the limit of detection of the plaque assay. Six of the inoculated bats had high CT values (36–39) in the lungs between days 3 and 7, corresponding to potential for low level infection or lung contamination from the inoculum (Table 1). Four bats in total were kept until day 28 post-infection to assess their serologic response; one of these bats had a PRNT90 antibody titer of 20, whereas the other three did not have neutralizing antibodies.

### 3.2. Histopathology

The histopathology findings are summarized in Appendix A. All the control bats died or were euthanized within 14 days of capture and did not thrive in captivity. Histologic findings in control bats included signs of poor intake, parasitism and bacteremia. Specific lesions included gastric erosions with bile staining (n = 3), ulcerative gastritis with or without intraluminal trematodes (n = 2), intestinal coccidiosis (n = 1), intestinal nematodiasis (n = 1), biliary trematodiasis (n = 1), colonic trematodiasis (n = 1) and mild tracheitis and pharyngitis (n = 1). Six treatment bats also showed signs of ill thrift and died prior to planned euthanasia time points. Histologic findings in these bats included gastric erosions with bile staining (n = 1), ulcerative gastritis with or without intraluminal trematodes (n = 2), intestinal coccidiosis (n = 1), intestinal trematodiasis (n = 1), splenic lymphoid depletion (n = 1), multifocal random hepatocellular necrosis (n = 1) and mucinous debris in nasal turbinates (n = 2). These lesions were interpreted as secondary to ill thrift and unrelated to SARS-CoV-2 infection.

Fourteen of the eighteen challenged bats had prominent bronchus-associated lymphoid tissue (BALT), versus two of six control bats. Seven of the eighteen treatment bats showed evidence of splenic lymphoid hyperplasia, versus zero of the six control bats. Three of the eighteen treatment bats showed evidence of erosive to ulcerative stomatitis, possibly due to repeated oral swabbing, versus zero of six controls.

### 3.3. Immunohistochemistry

Immunohistochemical staining did not result in the detection of any viral antigen in any bat tissues.

## 4. Discussion

SARS-CoV-2 continues to emerge as a zoonotic virus capable of infecting a variety of animal species, including cats; dogs; mink; various wild rodents, including deer mice; white-tailed deer; and non-human primates [17,19,20,21,22,23,24]. To date, two bat species have been evaluated experimentally for susceptibility to infection and capacity to shed virus, including big brown bats and Egyptian fruit bats [10,11]. While one of the contact Egyptian fruit bats became infected following exposure to inoculated animals, none of the experimentally infected big brown bats shed virus or transmitted, nor did bats of either species develop clinical signs of disease. However, there are more than 1400 bat species worldwide, and while some general assumptions can be made regarding species susceptibility, each species may respond differently to infection [25]. Therefore, it is important to acknowledge that not all bats are identical, and lack of infection in one species does not preclude resistance to infection in another. The goal of this study was to determine susceptibility of Brazilian free-tailed bats to SARS-CoV-2 infection, as these bats are widely distributed throughout the Americas, are migratory and are one of the most abundant species in the New World. The migratory nature of this species as it moves to new locations throughout its yearly cycle could increase its ability to spread a disease to a much wider range of hosts. In addition, the fact that this species is known to cohabitate with other bat species increases the risk of passing pathogens to them. This is important, as some *Myotis* bats have been shown to be more susceptible to white-nose syndrome when dealing with concurrent viral infection [26]. Free-tailed bats tend to roost in large numbers and are frequently found in close association with humans, making them more likely to come into contact with humans and, potentially, SARS-CoV-2. Individuals captured for this study were collected from an urban structure. Furthermore, based on the available data, including efficient ACE2 orthologue receptor binding and entry in vitro, *Tadarida* bats are perhaps one of the species most likely to be susceptible to the virus in North America, although to date, only alphacoronaviruses have been isolated from this species [14,15,27]. Importantly, bat rehabilitators and other people who work directly with these bats need to know the risk that we as humans pose to the bats, as well as the potential risk bats may pose to handlers. Of particular concern is the ability of SARS-CoV-2 to mutate rapidly in new hosts [28,29]; the threat of emergence of a new variant of concern following intraspecies transmission is a very serious problem to consider.

In this study, Brazilian free-tailed bats were found to be minimally permissive to infection with two strains of SARS-CoV-2. The first strain of SARS-CoV-2 isolated in the U.S., strain WA01, is the first strain bats may have been exposed to in the U.S., and at the time of these experiments, the Delta variant was the predominant circulating strain worldwide. Our results indicate that inoculation with either strain leads to low-level infection, as demonstrated by lack of infectious virus shedding orally or in the lungs, and high CT values in the oral swabs and lungs of some individuals, as might be expected from a transient and likely non-productive infection. Prominent bronchus-associated lymphoid tissue (BALT) and splenic lymphoid hyperplasia was seen in several bats, and may suggest a non-specific immune response, but no viral antigen was detected in lymphoid tissues via immunohistochemistry. One of four bats that were sampled after acute infection (>7 DPI) seroconverted, which indicates the potential for the virus to replicate sufficiently to elicit a specific antibody response, but because the antibody titer was low and was not seen in all bats, this does not indicate a robust immune response, and supports our conclusion that this species of bat is unlikely to serve as a wildlife reservoir SARS-CoV-2, at least for the two SARS-CoV-2 variants used in this study. Furthermore, the Delta variant, which is considered to be more virulent in other animal species, did not induce a different response in the bats compared to WA01 [30,31]. Overall, the lack of infectious viral shedding, lack of evidence of clinical disease, and minimal antibody response suggest that Brazilian free-tailed bats are minimally competent hosts for SARS-CoV-2 and are unlikely to serve as a reservoir for infection of bats or other species.

## 5. Conclusions

Wildlife likely played a role in the initial emergence of SARS-CoV-2 and could potentially serve as reservoir hosts in the future. At present, the burden of disease remains in humans, but the increasing evidence of circulation SARS-CoV-2 in white-tailed deer populations suggests that SARS-CoV-2 may become endemic in wildlife, which would significantly reduce our ability to eradicate the virus [32,33,34]. Of particular concern is the generation of new variants in a wildlife reservoir that may spill over once again into humans and lead to additional epidemics. To date, there has been no indication that this is the case, but scientists, wildlife managers and health officials must remain vigilant if we are to prevent such an emergence from occurring. Experimental investigations such as the one described herein are designed to inform about possible scenarios should a spillover from humans into animals occur, and based on this study, we believe that a spillover from humans to Brazilian free-tailed bats is unlikely. However, caution should still be practiced at all times when in contact with any wildlife. The COVID-19 pandemic has shown that the growing human–wildlife interface is increasingly exposing people and animals to each other’s pathogens, sometimes with dire consequences. Thus, while Brazilian free-tailed bats may not be especially at risk of SARS-CoV-2 infection, we cannot predict their response to new variants. As such, it is recommended that people interacting with any bat species or wild animals use appropriate personal protective equipment and take precautions to avoid spillover events in the future.

## Figures and Tables

**Table 1 viruses-14-01809-t001:** rt-PCR CT values for bats during acute (<7 day) infection.

Animal ID	SARS2 Strain	Day Died/Euthanized	Oral Swab PCR + Days	Lung CT Value
Bat 7	WA01	2	1–2	36.5
Bat 1	WA01	3	1–3	Undetermined
Bat 2	WA01	3	1–3	38.5
Bat 3	WA01	3	1–3	Undetermined
Bat 6	WA01	4	1–3	37.1
Bat 5	WA01	7	2–7	37
Bat 8	WA01	28	1–7	Not tested
Bat 9	WA01	28	1–7	Not tested
Bat 10	WA01	28	1–7	Not tested
Bat 4D	Delta	3	1–3	38.4
Bat 7D	Delta	3	1–3	Undetermined
Bat 8D	Delta	3	1–3	38.8
Bat 2D	Delta	7	1–7	Undetermined
Bat 5D	Delta	7	2–7	Undetermined
Bat 9D	Delta	7	1–7	Undetermined
Bat 3D	Delta	14	2–7	Not tested
Bat 6D	Delta	14	1–7	Not tested
Bat 1D	Delta	28	1–7	Not tested

## Data Availability

Not applicable.

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
