# Peer review of "Experimental Infection of Brazilian Free-Tailed Bats (Tadarida brasiliensis) with Two Strains of SARS-CoV-2"

_viruses, 2022, doi:10.3390/v14081809_

Round 1

Reviewer 1 Report

General comments

The manuscript by Angela M. Bosco-Lauth and colleagues described the experimental infection of two SARS-CoV-2 variants with Brazilian free-tailed bats. The manuscript is well written with very comprehensive methodology.

There are a few points that need further clarification.

1.       It seems that the bats used were not tested for Coronavirus prior to experiment.

2.       Please provide the age class and sex in the experiment Although the estimation was mentioned in the M&M, the information is missing.

3.       Oral swabs were collected daily for virus isolation only. Please clarify why they were not tested by PCR. Virus isolation maybe less sensitive than PCR assay.

4.       It is clear enough regarding the number of bats used in the experiment, those died or harvested. However, it is better to have a graph/table to document different time points (time of inoculation, time of bat died/harvested), number of bat died/harvested by virus strain.

5.       In 3.2. Histophathology section, it was stated “All the control bats died or were euthanized within 14 days…..”. What authors mean by “control bats”? The control bats were not mentioned in methods except those died before the inoculation.

6.       Similarly in the same section, “Fourteen of the eighteen treatment bats ……” What author mean by “treatment”? There is no information about that.

7.       Line 220, “Our results indicate….. leads to minimal infection….”. The ‘minimal infection’ is misleading and in contrast to the final conclusion that the bats used were not competent for SARS-CoV-2.

8.       Line 230, I propose to change to “unlikely to serve as a wildlife reservoir for the two SARS-CoV-2 variants used in the study”

9.       In the discussion, it would be interesting that the authors discuss about receptor of the Brazilian free-tailed bat susceptibility to SARS-CoV-2.

Reviewer 2 Report

In the manuscript titled with "Experimental infection of Brazilian free-tailed bats (Tadarida brasiliensis) with SARS-CoV-2", the authors described results from SARS-CoV-2 experimental infection in bats. Overall, the study is interesting, but the data can be improved. 

1, I understand the bat infection exp can be difficult, yet the authors are suggested to do more analysis on the viral positive bats to answer important virological questions. For example, the viral replication dynamics in this bat? Shedding? Pictures showing histological changes, compared to hamster? Any immune difference between bats and other mammals? The authors are also suggested to test Abs but not only nAbs. Also, is the bat ACE2 from this species susceptible to SARS-CoV-2?

Overall, I believe it is a good project with space for further improvement.
